# Development of the Cerebrospinal Fluid in Early Stage after Hemorrhage in the Central Nervous System

**DOI:** 10.3390/life11040300

**Published:** 2021-04-01

**Authors:** Petr Kelbich, Aleš Hejčl, Jan Krejsek, Tomáš Radovnický, Inka Matuchová, Jan Lodin, Jan Špička, Martin Sameš, Jan Procházka, Eva Hanuljaková, Petr Vachata

**Affiliations:** 1Biomedical Centre, Masaryk Hospital Ústí nad Labem, 401 13 Ústí nad Labem, Czech Republic; inka.matuchova@kzcr.eu (I.M.); eva.hanuljakova@kzcr.eu (E.H.); 2Department of Clinical Immunology and Allergology, Faculty of Medicine and University Hospital in Hradec Králové, Charles University in Prague, 500 03 Hradec Králové, Czech Republic; jan.krejsek@fnhk.cz; 3Laboratory for Cerebrospinal Fluid, Neuroimmunology, Pathology and Special Diagnostics Topelex, 190 00 Prague, Czech Republic; 4Department of Neurosurgery, Masaryk Hospital Ústí nad Labem, J. E. Purkinje University, 401 13 Ústí nad Labem, Czech Republic; ales.hejcl@kzcr.eu (A.H.); tomas.radovnicky@kzcr.eu (T.R.); jan.lodin@kzcr.eu (J.L.); martin.sames@kzcr.eu (M.S.); petr.vachata@kzcr.eu (P.V.); 5International Clinical Research Center, St. Anne’s University Hospital, 656 91 Brno, Czech Republic; 6Institute of Experimental Medicine, Academy of Sciences of the Czech Republic, 117 20 Prague, Czech Republic; 7Department of Clinical Biochemistry, Masaryk Hospital Ústí nad Labem, 401 13 Ústí nad Labem, Czech Republic; jan.spicka@kzcr.eu; 8Department of Neurosurgery, 2nd Faculty of Medicine, Charles University in Prague, 110 00 Prague, Czech Republic; 9Department of Anesthesiology, Perioperative Medicine and Intensive Care, Masaryk Hospital Ústí nad Labem, J. E. Purkinje University, 401 13 Ústí nad Labem, Czech Republic; jan.prochazka@kzcr.eu; 10Department of Neurosurgery, Faculty of Medicine and University Hospital in Pilsen, Charles University in Prague, 301 00 Pilsen, Czech Republic

**Keywords:** CNS haemorrhage, inflammation in CNS, cerebrospinal fluid, total protein in CSF, erythrocytes in CSF, neutrophils in CSF, coefficient of energy balance, aspartate aminotransferase in CSF

## Abstract

Extravasation of blood in the central nervous system (CNS) represents a very strong damaged associated molecular patterns (DAMP) which is followed by rapid inflammation and can participate in worse outcome of patients. We analyzed cerebrospinal fluid (CSF) from 139 patients after the CNS hemorrhage. We compared 109 survivors (Glasgow Outcome Score (GOS) 5-3) and 30 patients with poor outcomes (GOS 2-1). Statistical evaluations were performed using the Wilcoxon signed-rank test and the Mann–Whitney U test. Almost the same numbers of erythrocytes in both subgroups appeared in days 0–3 (*p* = 0.927) and a significant increase in patients with GOS 2-1 in days 7–10 after the hemorrhage (*p* = 0.004) revealed persistence of extravascular blood in the CNS as an adverse factor. We assess 43.3% of patients with GOS 2-1 and only 27.5% of patients with GOS 5-3 with low values of the coefficient of energy balance (KEB < 15.0) in days 0–3 after the hemorrhage as a trend to immediate intensive inflammation in the CNS of patients with poor outcomes. We consider significantly higher concentration of total protein of patients with GOS 2-1 in days 0–3 after hemorrhage (*p* = 0.008) as the evidence of immediate simultaneously manifested intensive inflammation, swelling of the brain and elevation of intracranial pressure.

## 1. Introduction

Hemorrhage in the central nervous system (CNS) is associated with a high risk of severe impairment of CNS or even the death [1,2,3]. Despite survive in the initial period patients are at the risk of severe complications such as microvessel constriction, large vessel vasospasm, re-bleeding or bacterial neuroinfection [2,3,4,5,6,7,8,9]. These patients are monitored at intensive care units including clinical evaluation, repeated computed tomography scans, invasive intracranial monitoring as well as repeated cerebrospinal fluid (CSF) analysis [10,11,12].

Hemorrhage in the CNS induces a local inflammatory response. Dumont et al. refer inflammation in response to subarachnoid blood as a plausible candidate pathway leading to cerebral vasospasm [2]. Wang reviews the roles of cytokines, proteases, and reactive oxygen species (ROS) in the intracranial hemorrhage (ICH) induced secondary brain injury. The special role in its pathogenesis is given to neutrophils [3]. Fassbender et al. found that the release of IL-1β, IL-6 and TNFα in the subarachnoidal space of patients with SAH is associated with the development of increased cerebral blood flow velocities in basal cerebral arteries. They observed that intrathecal secretion of proinflammatory cytokines after SAH is significantly increased in patients with poor clinical outcome [4]. Sercombe et al. refer that if the SAH is not immediately fatal, it may be the subject to the next complications, in particular hypertension and, later, cerebral vasospasms and re-bleeding. They also describe the contribution of the inflammatory reaction in the subarachnoid compartment to the development of vasospasms [5]. Tso and Macdonald refer that after SAH inflammatory mediators such as interleukin (IL)-1β, IL-6 and tumor necrosis factor α (TNFα), and oxidative damage from neutrophils and macrophages may result in direct damage to the microvasculature, resulting in damage to the blood–brain barrier (BBB), brain edema and brain injury [7]. Miller et al. describe the relationship between elevations of inflammatory mediators in the CSF, onset of vasospasm and decreased neurological outcomes of patients after SAH. They found that TNFα levels in poor-grade SAH patients were shown to be correlated with severity of vasospasm. They consider further the CSF concentrations of IL-6 as an early marker to predict vasospasm development [8]. In addition, hemolytic events after CNS hemorrhage result in large quantities of extracellular hemoglobin. Subsequently, as the heme-iron of hemoglobin is converted from ferrous to ferric form, hemoglobin is oxidized quickly releasing its heme moieties. Extracellular hemoglobin/heme/iron initiates a cascade of free radical-induced damage, oxidative stress and formation of damage associated molecular patterns (DAMP) is initiating the inflammatory response. The complexes hemoglobin–haptoglobin are formed. These complexes are recognized by CD163 scavenger receptors expressed especially on alternatively polarized M2 macrophages. Complexes are internalized and intracellularly subjected to the action of hemoxigenase to eliminate their proinflammatory and oxidative potentials [13].

Our previous effort was to detect predominantly purulent inflammatory complications of bacterial origin in the CNS after hemorrhage using basic analysis of the CSF [6]. This retrospective study is aimed at the comprehensive assessment of local inflammation in the CSF in the early stage after the CNS hemorrhage and its relationship to the clinical outcome. We compared results of the CSF analysis of survivors (Glasgow Outcome Score [GOS] 5–3) and patients with poor outcome (GOS 2-1) on days 0–3 and 7–10 after the attack of hemorrhage.

### 1.1. Cytological Investigation of the CSF

The typical presentation of hemorrhage occurring within the CSF are erythrophages with freshly phagocytosed or decolored erythrocytes, siderophages and hematoidin crystals [14,15,16,17,18,19].

The presence and composition of immunocompetent cells in the CSF was evaluated in our study. The presence of lymphocytes and monocytes is usually indicating slight serous inflammation which is typical for the posthemorrhagic clean-up reaction. The predominant presence of innate immunity cells, such as neutrophils, is indicating the inflammatory reaction to adverse consequences of hemorrhage within the CNS, especially vasospasms, re-bleeding and bacterial neuroinvasion [2,3,5,14,18,19,20].

### 1.2. Investigation of Biochemical Parameters in the CSF

We used parameters of basic biochemical analysis of the CSF. These are concentrations of total protein, energy parameters and a measurement of catalytic activities of aspartate aminotransferase (AST) in the CSF [18,19,20,21,22,23]. Their advantages are common availability, reliable analysis and immediate response at low price.

Concentration of the total protein in the CSF usually correlates with permeability of the blood-cerebrospinal fluid barrier. Nevertheless, high concentrations of the total protein in the CSF of patients after the CNS hemorrhage is very often influenced by impairment of the CSF circulation [18,24].

The AST catalytic activity in the CSF is reliable and easily accessible parameter of the CNS tissue injury [22].

#### Energy Assessment of Inflammation in the CSF

To address the presence and the extent of the immune reactions within the CSF compartment, it is essential to analyze functional as well as morphological parameters within the CSF. Activated immunocompetent cells display high energy requirements [25,26,27,28]. In order to evaluate the extent of immune system activation within the CSF compartment, measurement of energy parameters is performed. Many authors use molar concentrations of glucose or lactate in the CSF [9,29,30,31,32,33,34,35,36]. However, value of these parameters is limited. Concentrations of glucose in the CSF are dependent on the concentrations of blood glucose. Concentrations of lactate in the CSF are not only reflecting the extent of anaerobic metabolism in the CSF, but are also dependent on the supply of energy substrate (glucose). Therefore we derived an equation to identify the theoretical average number of molecules of adenosine triphosphate (ATP) produced from one molecule of glucose under set conditions in the CSF compartment. We call this parameter the coefficient of energy balance (KEB; in Czech Koeficient Energetické Bilance) [21,22,37]:(1)$$\mathrm{KEB}\;=\;38-18\frac{\left[lactate\right]}{\left[glucose\right]}$$ Legend:
[glucose] = molar concentration of glucose in the CSF (mMol·L^−1^).[lactate] = molar concentration of lactate in the CSF (mMol·L^−1^).

Normal energy condition in the CSF compartment is characterized with the high KEB values over 28.0. Activation of immunocompetent cells caused an increase of anaerobic metabolism which is presented by the decrease of KEB values under 28.0. The lowest KEB values (KEB < 10.0) are typical for very intensive inflammation with oxidative burst of professional phagocytes (neutrophils and macrophages) and ROS production–e.g., in the case of purulent inflammation [22,37].

## 2. Material and Methods

### 2.1. Patients

We selected 139 patients with two analyses of their CSF in this retrospective study. The first analysis was done in between the 0 to 3 day and the second one in between the 7th to 10th day after the CNS hemorrhage. On days 0–3 the perioperative collections of the CSF were completed with the purpose of cranial decompression, improvement of surgical conditions and gaining of input information about the CSF compartment. All takings were made using lumbar puncture or drainage.

The patients were clinically evaluated during hospital discharge using the Glasgow Outcome Score (GOS). We divided these patients into two subgroups—109 survivors (GOS 5-3) and 30 patients with poor outcome (GOS 2-1) (Table 1).

Furthermore, we present CSF findings in two considerably different control groups of patients—500 patients with normal CSF findings and 109 patients with purulent inflammation in the CNS.

### 2.2. CSF Analysis

CSF samples were repetitively obtained via external CSF drainage into a test tube without anticoagulation agents and immediately transported for laboratory examination. Immediately after receiving the sample, we evaluated the cell count and permanent cytological smear developed using a cyto-centrifuge method was made. Another part of the sample was centrifuged (10 min; 1500× *g*) and the concentrations of total protein, glucose, lactate and AST catalytic activities were determined. The rest of the supernatant was temporarily stored in refrigerator at +4 °C to +8 °C for potential future analysis.

In each case, we calculated the total number of elements in the CSF using a Fuchs-Rosenthal chamber and evaluated smear stained with Hemacolor (Merck Co., Darmstadt, Germany). Olympus BX40 microscope (Olympus, Tokio, Japan) was used.

We analyzed molar concentrations of glucose in the CSF using the hexokinase method, molar concentrations of lactate in the CSF using the lactate–oxidase and peroxidase, mass concentrations of total protein in the CSF using the turbidimetric method with benzetonium chloride and catalytic activities of AST in the CSF using the IFCC (International Federation of Clinical Chemistry) method with a Cobas 6000 analyzer (Roche Co., Basel, Switzerland). We calculated the KEB for each case [22,37].

### 2.3. Statistical Analysis

Concentrations of total protein, glucose and lactate, KEB values, numbers of nucleated cells and erythrocytes, the percentage of neutrophils and AST catalytic activities in the CSF are presented in Table 2 and Table 3 as a median and the 1st and the 3rd quartile. Firstly, we tested differences between these parameters on days 0–3 and 7–10 after the CNS hemorrhage in both subgroups of patients using the Wilcoxon signed-rank test (Table 2). Secondly, we tested differences between these parameters in both subgroups of patients on days 0–3 and 7–10 after the CNS hemorrhage using the Mann–Whitney U test (Table 3). All statistical tests were done via Statistica 13.3 software (StatSoft Inc., Tulsa, OK, USA) and *p* values < 0.05 were considered as significant.

## 3. Results

Table 2 and Table 3 present the comparisons between CSF parameters of survivors (GOS 5-3) and patients with poor outcome (GOS 2-1) immediately (days 0–3) and later (days 7–10) after the CNS hemorrhage. In addition, the Table 2 contents data of two control subgroups of patients—with normal CSF findings and with purulent inflammation in the CNS (Table 2).

Immediately after the CNS hemorrhage the numbers of erythrocytes in the CSF were almost the same in survivors and patients with poor outcome. Significant differences between these subgroups were apparent in days 7–10. While number of erythrocytes in the CSF of survivors significantly decreased, their number in patients with poor outcome was insignificantly higher (Table 2 and Table 3).

The increase in the number of leukocytes in the CSF on days 0–3 and on days 7–10 after the CNS hemorrhage in both subgroups of patients was approximately the same. It is significant in survivors and insignificant in patients with poor outcome (Table 2 and Table 3).

Relative count of neutrophils in the CSF of both subgroups of patients on days 0–3 and 7–10 after the CNS hemorrhage were approximately the same (Table 2 and Table 3).

The elevation of total protein in the CSF immediately after the CNS hemorrhage was significantly higher in patients with poor outcome compared to survivors. On days 7–10 their values were approximately the same. The decrease of concentrations of total protein in the CSF during this time was significant in both subgroups of patients (Table 2 and Table 3).

Significantly higher concentrations of glucose in the CSF were found in patients with poor outcome compared to survivors on days 0–3 and 7–10 after the CNS hemorrhage. They significantly decreased in time in both subgroups of our patients (Table 2 and Table 3).

Immediately after the CNS hemorrhage the concentrations of lactate in the CSF were significantly higher in patients with poor outcome compared to survivors. They significantly decreased over time to similar values on days 7–10 after the CNS hemorrhage (Table 2 and Table 3).

Decreased KEB values in the CSF presented the higher extent of anaerobic metabolism in the CSF compartment in survivors and patients with poor outcome. They were neither significantly different in both subgroups of patients nor in time after the CNS hemorrhage (Table 2 and Table 3).

Immediately after the CNS hemorrhage AST catalytic activities in the CSF were similar in survivors and patients with poor outcome. Their significant increase was apparent in both subgroups between the days 0 and 3 and the days 7 and 10 after the CNS hemorrhage. The growth of AST catalytic activities was higher in patients with poor outcome (Table 2 and Table 3).

We found apparently more causes with low KEB values (<15.0) and apparently less causes with moderate KEB values (15.0–28.0) in patients with poor outcome compared with survivors immediately after the CNS hemorrhage (Figure 1).

The differences between frequencies of patients of both subgroups divided in accordance with their KEB values were not significant on days 7–10 after the hemorrhage (Figure 2).

While frequencies of survivors and patients with poor outcome with AST catalytic activities more than 30.0 IU/L in the CSF immediately after the CNS hemorrhage were almost the same, on days 7–10 frequency of patients with poor outcome was significantly higher (Figure 3).

## 4. Discussion

The CNS hemorrhage is a very serious brain injury with a high risk of persistent neurology deficits or even the death. Many variables can influence the outcome of patients. We focused on the CSF monitoring of these patients over several days and divided them into two subgroups in accordance with the severity of their impairment—survivors (GOS 5-3) and patients with poor outcomes (GOS 2-1). All these patients were followed during the first 3 days and on the second occasion between the 7th to 10th day after the CNS hemorrhage.

Some authors recognize the amount of bleeding in the CNS as one of the important factors of its prognosis [3,7]. In the contrary to this finding, we found similar numbers of erythrocytes in the CSF of both subgroups of our patients during the first three days after the CNS hemorrhage (Table 2). Significant differences between survivors and patients with poor outcome were only found in days 7 to 10 (Table 3). Whereas the count of erythrocytes significantly decreased in survivors, in patients with a poor outcome, numbers of erythrocytes did not decline. We concluded that this was due to a persistence of bleeding, a gradual release of large amounts of extravascular erythrocytes from the CNS into the CSF or re-bleeding in patients with a poor outcome [5]. On the other hand, relatively small amounts of extravascularly localized erythrocytes in the CNS of survivors were eliminated by the clean-up reaction effectively [15,16,17,18,19].

CNS hemorrhage and its consequent complications are evoking damaging local inflammatory response in the CNS [2,3,4,5,6,7,8,9]. Increased numbers of leukocytes, higher relative number of neutrophils and decreased KEB values in the CSF of survivors and patients with poor outcome are the evidences of local inflammatory response in the CNS after a hemorrhage in both subgroups of patients (Table 2 and Table 3) [3,21,22,37]. We compared the extent of inflammation in the CSF compartment between these subgroups using the cytological-energy investigation of the CSF. Immediately after the CNS hemorrhage the numbers of leukocytes in the CSF did not differ when compare both groups of patients (Table 3). Their next elevation in days 7 to 10 was also very similar in all patients (Table 1). In addition, we did not find any differences in the relative number of neutrophils in the CSF in both subgroups (Table 2 and Table 3). Therefore, it is not possible to consider the presence of immunocompetent cells in the CSF after the CNS hemorrhage as a suitable biomarker to predict the clinical outcome of patients.

Slightly decreased KEB values in patients with GOS 2-1 compared to patients with GOS 5-3 immediately after the CNS hemorrhage represents a trend to higher intensity of inflammation in patients with poor outcome. In addition, we found 43.3% of patients with poor outcome and only 27.5% survivors with very low KEB values under 15.0 (Figure 1). Anaerobic metabolism in the CSF compartment with KEB values under 15.0 usually reflects very intensive inflammatory complications with oxidative burst of professional phagocytes [22,37].

Elevation of total protein in the CSF immediately after the CNS hemorrhage is simultaneously influenced by many variables—amount of extravascular blood, inflammatory response and disorder of the CSF circulation to name some of them. It seems that the inflammatory reaction in the CNS immediately after the hemorrhage plays the key role. Inflammation is accompanied by a higher increase in the permeability of the blood-brain barrier (BBB) and presence of proinflammatory mediators in the CNS. It results in swelling of the brain, increase of intracranial pressure (ICP) and impairment of blood circulation in the brain and circulation of the CSF. In addition, inflammation in the CNS increases risk of vasospasms and secondary cerebral ischemia [2,3,4,5,7,8,24].

Elevation of total protein in the CSF is found in both subgroups of our patients. It is significantly higher in patients with poor outcome (Table 3). The concentrations of total protein in the CSF decline fast on the similar values in both subgroups during the first week after the CNS hemorrhage (Table 2 and Table 3). We conclude that the high concentration of total protein in the CSF of patients with poor outcome immediately after the CNS hemorrhage, together with a high amount of extravascular blood, represent the trend to intensive inflammatory reaction, swelling of the brain tissues and disorder of the CSF circulation.

Concentrations of glucose in the CSF of patients with poor outcomes compared with survivors were significantly higher in days 0 to 3 after the CNS hemorrhage (Table 3). Significant decline of glucose concentrations in the CSF in both subgroups of our patients was observed in days 7 to 10 (Table 2) probably reflecting the systemic changes in glucose metabolism. Our findings are in accord with the study of Matthew et al. who found hyperglycemia as a significant factor of poor outcome after aneurysmal SAH [38]. Similar dynamic of lactate concentrations in the CSF of our patients reflects the changes of glucose concentrations in relatively stabilized energy conditions in the CSF compartment (Table 2 and Table 3) [39,40].

AST catalytic activity in the CSF is recognized as a useful parameter to detect CNS tissue injury. However, its values in survivors and patients with poor outcome were not significantly different immediately after the CNS hemorrhage (Table 2). Only a few days later their elevation in the CSF of patients with poor outcomes was higher compared with survivors and represented significant progression of irreversible damage of the CNS tissue of these patients (Table 2 and Figure 3). These results are not surprising, because they are in accord with the conclusions of our recent study [23].

## 5. Conclusions

Extravasation of blood in the CNS parenchyma represents a very strong DAMP (damaged associated molecular patterns) which is followed by rapid inflammatory response and other pathologies which can cause worse outcome of patients.

The battery of our tests to monitor the CSF in patients after a CNS hemorrhage comprises the absolute count of leukocytes and erythrocytes, the differential cell count, concentrations of total protein, glucose and lactate, the calculation of the KEB values and catalytic activities of AST. We compared these parameters in two subgroups of patients after a CNS hemorrhage—in survivors (GOS 5-3) and in patients with poor outcome (GOS 2-1).

We found two key adverse prognostic factors. The first one is the persistence of bleeding or re-bleeding in the CNS. The second ones are the low KEB values (<15.0) and increased total protein concentration in the CSF both representing trend to the very intensive early inflammatory response in the CNS, brain swelling, elevation of intracranial pressure and impairment of the CSF circulation. Subsequently increased AST catalytic activities in the CSF of patients with poor outcome reveal more profound structural tissue injury in their CNS.

## Figures and Tables

**Figure 1 life-11-00300-f001:**
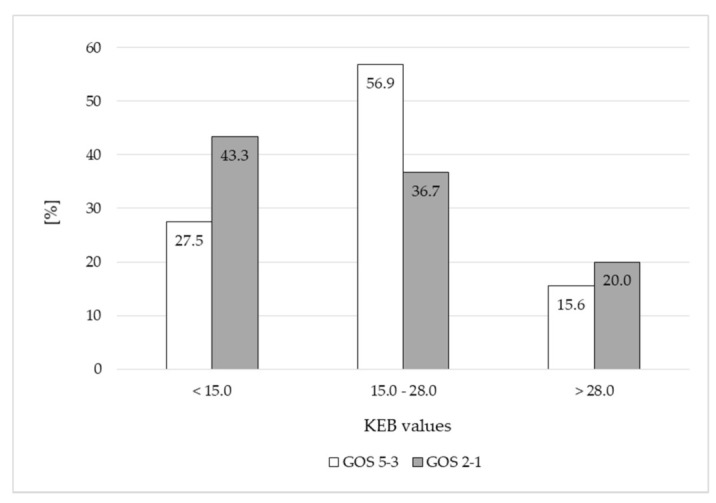
Frequencies of patients divided in accordance with KEB values in 0–3 days after the CNS hemorrhage.

**Figure 2 life-11-00300-f002:**
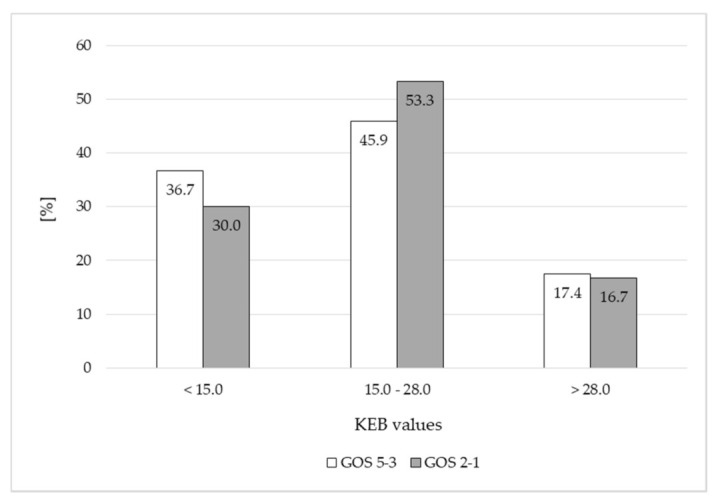
Frequencies of patients divided in accordance with KEB values in 7–10 days after the CNS hemorrhage.

**Figure 3 life-11-00300-f003:**
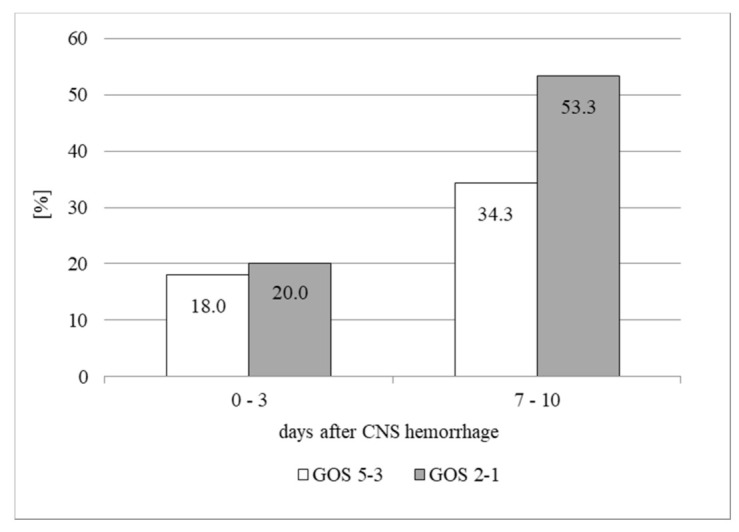
Frequencies of patients with AST catalytic activities >30.0 IU/L in the CSF in days 0–3 and 7–10 after the CNS hemorrhage.

**Table 1 life-11-00300-t001:** Distribution of our patients in accordance with outcome, the type of the CNS hemorrhage and demographic data.

Groups of Patients	GOS 5-3		GOS 2-1	
**Sex**	**Females**	**Males**	**Females**	**Males**
number	66	43	12	18	
median of age	61	60	73	64	
minimal age	25	33	44	44	
maximal age	88	84	83	77	
SAH	52	25	8	7	
ICH	10	18	3	9	
SAH + ICH	4	0	1	2	

Legend: GOS—Glasgow Outcome Score; SAH—number of causes with subarachnoid hemorrhage; ICH—number of causes with intracerebral hemorrhage.

**Table 2 life-11-00300-t002:** Investigation of the CSF of patients after the CNS hemorrhage—comparison between days 0 and 3 and 7 and 10 after the CNS hemorrhage; and two control groups of patients with normal findings and with purulent inflammation in the CNS.

Median (1st to 3rd Quartile)	Control Group 1 *n* = 500	Control Group 2 *n* = 109	GOS 5−3 *n* = 109		GOS 2−1 *n* = 30	
**CSF Parameters**	**Normal Findings**	**Purulent Inflammation** **in the Central Nervous System (CNS)**	**Day** **0–3**	**Day** **7–10**	**Day** **0–3**	**Day** **7–10**
**Erythrocytes** (elements/1 μL)	**0** (0–1)	**160** (21–843)	*p* = 0.002 *		*p* = 0.688	
			**26,112** (7680–98,133)	**13,867** (2475–48,640)	**24,747** (6272–165,973)	**40,875** (12,885–144,768)
**Leukocytes** (elements/1 μL)	**1** (1–2)	**2513** (800–6783)	*p* = 0.035 *		*p* = 0.068	
			**47** (12–381)	**141** (37–523)	**41** (5–193)	**209** (60–700)
**Neutrophils** (%)	**0** (0–0)	**83** (73–90)	*p* = 0.377		*p* = 0.681	
			**46** (26–66)	**54** (36–67)	**50** (27–76)	**47** (36–69)
**Total protein** (mg/L)	**296.0** (243.75–354.0)	**3021.5** (1596.75–5416.0)	p < 0.001 *		*p* = 0.002 *	
			**1625.0** (590.0–3990.0)	**654.0** (378.0–1047.0)	**4049.5** (1308.25–10,775.0)	**887.5** (384.75–1838.25)
**Glucose** (mmol/L)	**3.41** (3.19–3.65)	**0.50** (0.03–1.80)	p < 0.001 *		*p* = 0.002 *	
			**4.60** (3.84–5.70)	**3.54** (2.80–4.58)	**5.77** (4.87–6.89)	**4.24** (3.43–5.41)
**Lactate** (mmol/L)	**1.49** (1.35–1.60)	**10.43** (6.77–13.36)	*p* = 0.001 *		*p* = 0.002 *	
			**4.55** (3.29–6.15)	**3.62** (2.87–5.23)	**6.84** (4.60–8.72)	**4.23** (3.23–5.74)
**KEB**	**30.23** (29.55–30.94)	**−369.88** (−6184 to −36.98)	*p* = 0.130		*p* = 0.586	
			**21.16** (12.43–26.60)	**20.79** (5.82–27.30)	**17.48** (6.87–25.07)	**20.67** (10.41–25.08)
**AST** (IU/L)	**12.6** (10.2–15.0)	**21.6** (16.8–46.2)	*p* < 0.001 *		*p* = 0.001 *	
			**12.0** (6.6–24.0)	**24.6** (13.8–37.2)	**16.8** (9.6–25.2)	**31.8** (24.0–63.0)

Legend: GOS: Glasgow Outcome Score; *n*: number of patients; KEB: coefficient of energy balance; AST: aspartate aminotransferase; *: statistically significant (*p* < 0.05)—tested using by the Wilcoxon signed−rank test.

**Table 3 life-11-00300-t003:** Investigation of the cerebrospinal fluid (CSF) in two subgroups of patients after the CNS hemorrhage.

Median (1st to 3rd Quartile)	Day 0–3		Day 7–10	
**CSF Parameters**	**GOS 5-3** *n* = 109	**GOS 2-1** *n* = 30	**GOS 5-3** *n* = 109	**GOS 2-1** *n* = 30
**Erythrocytes** (elements/1 μL)	*p* = 0.927		*p* = 0.004 *	
	**26,112** (7680–98,133)	**24,747** (6272–165,973)	**13,867** (2475–48,640)	**40,875** (12,885–144,768)
**Leukocytes** (elements/1 μL)	*p* = 0.312		*p* = 0.747	
	**47** (12–381)	**41** (5–193)	**141** (37–523)	**209**( 60–700)
**Neutrophils** (%)	*p* = 0.547		*p* = 0.947	
	**46** (26–66)	**50** (27–76)	**54** (36–67)	**47** (36–69)
**Total protein** (mg/L)	*p* = 0.008 *		*p* = 0.126	
	**1625.0** (590.0–3990.0)	**4049.5** (1308.25–10,775.0)	**654.0** (378.0–1047.0)	**887.5** (384.75–1838.25)
**Glucose** (mmol/L)	*p* = 0.003 *		*p* = 0.048 *	
	**4.60** (3.84–5.70)	**5.77** (4.87–6.89)	**3.54** (2.80–4.58)	**4.24** (3.43–5.41)
**Lactate** (mmol/L)	p < 0.001 *		*p* = 0.097	
	**4.55** (3.29–6.15)	**6.84** (4.60–8.72)	**3.62** (2.87–5.23)	**4.23** (3.23–5.74)
**KEB**	*p* = 0.317		*p* = 0.937	
	**21.16** (12.43–26.60)	**17.48** (6.87–25.07)	**20.79** (5.82–27.30)	**20.67** (10.41–25.08)
**AST** (IU/L)	*p* = 0.242		*p* = 0.047 *	
	**12.0** (6.6–24.0)	**16.8** (9.6–25.2)	**24.6** (13.8–37.2)	**31.8** (24.0–63.0)

Legend: GOS: Glasgow Outcome Score; *n*: number of patients; KEB: coefficient of energy balance; AST: aspartate aminotransferase; *: statistically significant (*p* < 0.05)—tested using by the Mann-Whitney U test.

## Data Availability

All relevant data are within the paper.

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
