# Peer review of "Development of the Cerebrospinal Fluid in Early Stage after Hemorrhage in the Central Nervous System"

_life, 2021, doi:10.3390/life11040300_

Round 1

Reviewer 1 Report

The analysis conducted by the authors in the article entitled “Development of the cerebrospinal fluid in early stage after haemorrhage in the central nervous system” aimed the comprehensive assessment of local inflammation in the cerebrospinal fluid, based on the routine CSF laboratory examination, in the early stage after the CNS haemorrhage and its relationship to the clinical outcome.

Authors should explain, why CSF was collected from patients with CSN hemorrhage on days 0-3? If the hemorrhage is present on MRI or CT scan, the is no need for CSF collection for diagnostic purposes. Moreover, it is not stated in the article what was the way of the CSF collection (lumbar puncture? or via the intracranial shunt?).

What was the time and g force of the CSF centrifugation for biochemical testing?

Instead of “relative frequencies of neutrophils” should be “the percentage of neutrophils”.

Why authors give the unit [elements/3 μL] for erythrocytes and leukocytes? In routine laboratory diagnostics we use the unit [elements/μL]. Did the authors use some other way of result calculation? What was the method used for erythrocyte calculation? According to my experience there is no possibility to calculate erythrocyte count with the use of Fuchs-Rosenthal chamber or cytological smear.

CSF results presented in Table 2 and 3 are the same; they differ regarding only the p-values. Authors should think about another way of presenting the data or move one of the Table to supplementary data.

The percentage of neutrophils should be present with no decimal points, e.g. 83% instead of 82.7%. According to my experience when we count cells under the microscope, we do not divide them into  parts.

Authors should rewrite the following sentence in the Conclusion section: “Extravasation of blood in the CNS parenchyma represents a very strong DAMP (damaged associated molecular patterns) which is followed by rapid inflammatory response and other pathologies which can cause worse outcome of patients.”. They did not evaluate DAMPs in their study, thus in my opinion they should not mention these molecules in the Conclusion section. Authors should also try to indicate what is the practical application of their study in routine clinical practice.

Author Response

The analysis conducted by the authors in the article entitled “Development of the cerebrospinal fluid in early stage after haemorrhage in the central nervous system” aimed the comprehensive assessment of local inflammation in the cerebrospinal fluid, based on the routine CSF laboratory examination, in the early stage after the CNS haemorrhage and its relationship to the clinical outcome.

Authors should explain, why CSF was collected from patients with CSN hemorrhage on days 0-3? If the hemorrhage is present on MRI or CT scan, the is no need for CSF collection for diagnostic purposes. Moreover, it is not stated in the article what was the way of the CSF collection (lumbar puncture? or via the intracranial shunt?).

My response:

On days 0-3 the perioperative collections of the CSF were done with the purpose of cranial decompression, improvement of surgical conditions and gaining of input information about the CSF compartment. All takings were made using lumbar puncture or drainage.

I have added this information to the manuscript (see “Material and Methods” – the chapter 2.1. “Patients”).

What was the time and g force of the CSF centrifugation for biochemical testing?

My response:

It is 10 minutes and 1,500 g. I have added this information into the text (see “Material and Methods” – the chapter 2.2. “CSF analysis”).

Instead of “relative frequencies of neutrophils” should be “the percentage of neutrophils”.

My response:

I have changed “relative frequencies of neutrophils” to “the percentage of neutrophils”.

Why authors give the unit [elements/3 μL] for erythrocytes and leukocytes? In routine laboratory diagnostics we use the unit [elements/μL]. Did the authors use some other way of result calculation? What was the method used for erythrocyte calculation? According to my experience there is no possibility to calculate erythrocyte count with the use of Fuchs-Rosenthal chamber or cytological smear.

My response:

25 years ago when I started with investigation of the CSF numbers of elements were presented as “elements/3 μL”. It is true, that currently many laboratories prefer “elements/1 μL”. I respect this progressive way and I recalculated all results in accordance with your recommendation.

CSF results presented in Table 2 and 3 are the same; they differ regarding only the p-values. Authors should think about another way of presenting the data or move one of the Table to supplementary data.

My response:

I understand your objection. It is true that the presented values are the same in both tables. Contrary to this fact the aims of these tables are different. Table 2 assesses the development of the CSF in both subgroups of patients in time using the Wilcoxon signed-rank test. The aim of table 3 is to compare the CSF between patients with GOS 5-3 and GOS 2-1 in days 0-3 and 7-10 using the Mann-Whitney U test. It means that the presented information in tables 2 and 3 are significantly different. I thought about the creation of only one table. Unfortunately I did not find any suitable way without the loss of intelligibility.

The percentage of neutrophils should be present with no decimal points, e.g. 83% instead of 82.7%. According to my experience when we count cells under the microscope, we do not divide them into  parts.

My response:

I have removed decimal points in both tables.

Authors should rewrite the following sentence in the Conclusion section: “Extravasation of blood in the CNS parenchyma represents a very strong DAMP (damaged associated molecular patterns) which is followed by rapid inflammatory response and other pathologies which can cause worse outcome of patients.”. They did not evaluate DAMPs in their study, thus in my opinion they should not mention these molecules in the Conclusion section. Authors should also try to indicate what is the practical application of their study in routine clinical practice.

My response:

I am an immunologist and a biochemist. My aim is to observe and to evaluate immune and biochemical processes which result in serious clinical conditions. Interaction between molecular structures DAMP released from erythrocytes and pattern recognition receptors of professional phagocytes is the “big bang” of local inflammatory response in the CNS immediately after the haemorrhage. We consider the practical application of this knowledge as a perspective way to the improvement of prediction of the brain injury and earlier and more efficient therapeutic intervention.

Reviewer 2 Report

The manuscript is well written and the topic is of clinical importance. However, I have the following comments:

24: Days 7-10 was the longest available follow-up time? It is quiet short…

32: Did you also measure the CSF/serum albumin quotient (Qalb) which is considered to be a better indicator of a CSF-blood barrier function?

42: …and cranial ultrasound I assume?

96: The CSF/serum albumin quotient (Qalb) has been demonstrated to correlate better with the blood-cerebrospinal fluid barrier. Did you also measure Qalb?

146: Did you have any CT negative patients with SAH?

147: Is it not a rather a prospective approach when you measured CSF at specific time points including measuring of AST which I suppose is not a standard parameter?

159: What are the patients’ diagnoses in control group 2? Did they all have bacterial meningitis?

167: I would suggest also measuring serum concentrations of glucose and AST and putting it in relation to CSF concentrations. Although most of the CSF lactate is supposed to be from anaerobic CNS metabolism, there is also a carrier depend transport mechanisms for systemic produced lactate which is normally negligible. However, did your intensive care patients have extreme elevate serum lactate concentrations? Did you check?

274: Did brain imaging confirm a persisting bleeding?

299: It has been demonstrated that the amount of blood in CSF influences the measured CSF protein concentrations: “The Influence of Blood Contamination on Cerebrospinal Fluid Diagnostics” Front Neurol . 2019 Jun 12;10:584. doi: 10.3389/fneur.2019.00584. eCollection 2019. I would suggest considering this work for your discussion.

Author Response

Dear reviewer,

Thank you very much for proofreading of our manuscript and your comments and recommendations.

The manuscript is well written and the topic is of clinical importance. However, I have the following comments:

24: Days 7-10 was the longest available follow-up time? It is quiet short…

My response:

The former design of this study calculated with 3 weeks long period. Finally we have changed our purpose and we evaluated the CSF immediately after the CNS haemorrhage (day 0-3) and few days later (day 7-10). The reason is simple – sufficient number of patients for serious evaluation. We are confident that this period is satisfactory for evaluation of the CSF changes in relationship of vasospasms.

32: Did you also measure the CSF/serum albumin quotient (Qalb) which is considered to be a better indicator of a CSF-blood barrier function?

My response:

I agree, that Qalb. is more accuracy parameter for evaluation of permeability of blood-cerebrospinal fluid barrier and also disorder of CSF circulation. We standard use this parameter for evaluation of gentle differences in the CSF of neurological patients. On the other hand, we consider the concentrations of total protein in considerably alterated CSF samples of serious neurosurgical patients as sufficient.

42: …and cranial ultrasound I assume?

My response:

All our patients after the CNS haemorrhage are standardly monitored using the Transcranial Doppler Tomography every day from surgery to the 14th day.

96: The CSF/serum albumin quotient (Qalb) has been demonstrated to correlate better with the blood-cerebrospinal fluid barrier. Did you also measure Qalb?

My response:

See above.

146: Did you have any CT negative patients with SAH?

My response:

Yes, we also have patients with SAH with negative CT scan in our practice.

147: Is it not a rather a prospective approach when you measured CSF at specific time points including measuring of AST which I suppose is not a standard parameter?

My response:

We have used AST catalytic activities as a standard CSF parameter since 1996 (see our paper “Can Aspartate Aminotransferase in the Cerebrospinal Fluid Be a Reliable Predictive Parameter? Brain Sci. 2020, 10, 698; doi:10.3390/brainsci10100698”).

159: What are the patients’ diagnoses in control group 2? Did they all have bacterial meningitis?

My response:

Yes, all these patients had bacterial purulent inflammation of the CNS.

167: I would suggest also measuring serum concentrations of glucose and AST and putting it in relation to CSF concentrations. Although most of the CSF lactate is supposed to be from anaerobic CNS metabolism, there is also a carrier depend transport mechanisms for systemic produced lactate which is normally negligible. However, did your intensive care patients have extreme elevate serum lactate concentrations? Did you check?

My response:

We evaluated energy parameters (glucose, Qglu., lactate and KEB) and AST catalytic activities in our former studies (for example: Coefficient of energy balance, a new parameter for basic investigation of the cerebrospinal fluid. Clin Chem Lab Med 2014, 52, 1009-1017. DOI 10.1515/cclm-2013-0953; Can Aspartate Aminotransferase in the Cerebrospinal Fluid Be a Reliable Predictive Parameter?” Brain Sci. 2020, 10, 698; doi:10.3390/brainsci10100698). Their conclusions are essential for our current work.

Management of urgent therapy in intensive care is difficult and we do not want to make it more complicated. Valid parallel analysis of glucose and lactate also demands special approach. Taking of blood must be done to the special test tube with Na+ fluoride (for inhibition of glycolysis) and the CSF must be collect 20 minutes after the blood collection. Therefore we did not commonly check lactate concentrations in blood and now we have not data for our retrospective study. Complex monitoring of our patients (including the cytological-energy analysis of the CSF) allows us to distinguish relatively rare cases of systemic hypoxia which influence the energy changes in the CSF compartment.

274: Did brain imaging confirm a persisting bleeding?

My response:

Persistence of bleeding was not confirmed by imaging. All cases were detected using the CSF analysis.

299: It has been demonstrated that the amount of blood in CSF influences the measured CSF protein concentrations: “The Influence of Blood Contamination on Cerebrospinal Fluid Diagnostics” Front Neurol . 2019 Jun 12;10:584. doi: 10.3389/fneur.2019.00584. eCollection 2019. I would suggest considering this work for your discussion.

My response:

Thank you very much for recommendation of this article. Presented experimental data are valuable for my practice and for my publication activities. I am confident, that CSF contamination by blood is not crucial problem in detection of acute bleeding in the CNS of neurosurgical patients. Changes of their CSF in vivo are in these cases usually massive and unmistakable. On the other hand, the recommended publication will help me especially in evaluation of typical neurological patients. So, I cite recommended paper in our manuscript (see reference no. 19).

Yours sincerely,

Petr Kelbich et al.

Reviewer 3 Report

The paper is good, although the fact that persistent or relapsing CNS hemorrhage results in a worse outcome is well known. Some points raised:
(1) The authors should clearly report the type of CNS hemorrhage in their patients. Was it subarachnoid hemorrhage, intraparenchymal hemorrhage (hemorrhagic stroke) or both?
(2) In tables 2 and 3: I believe that some numbers are not correctly presented. For example in table 3, day 7-10 erythrocytes in GOS 5-3 are 41 and 600 (2 numbers)? I believe that it is one number 41,600, but the authors should clarify and/or correct.
(3) In fig 5 (or corresponding part of the text) I couldn’t see a P value.

Author Response

Dear reviewer,

Thank you very much for proofreading of our manuscript and your comments and recommendations.

The paper is good, although the fact that persistent or relapsing CNS hemorrhage results in a worse outcome is well known.

My response:

I absolutely agree. I am glad that this generally known fact is also visible in our study.

Some points raised:
(1) The authors should clearly report the type of CNS hemorrhage in their patients. Was it subarachnoid hemorrhage, intraparenchymal hemorrhage (hemorrhagic stroke) or both?

My response:

These information are presented in the table 1, including differentiation to females and males.

(2) In tables 2 and 3: I believe that some numbers are not correctly presented. For example in table 3, day 7-10 erythrocytes in GOS 5-3 are 41 and 600 (2 numbers)? I believe that it is one number 41,600, but the authors should clarify and/or correct.

My response:

I have corrected numbers in both tables.

(3) In fig 5 (or corresponding part of the text) I couldn’t see a P value.

My response:

There was no calculated p value. This graph only presents percentages of cases with AST catalytic activities over 30.0 IU/L in both subgroups of patients in both evaluated periods.

Yours sincerely,

Petr Kelbich et al.

Round 2

Reviewer 2 Report

The authors have satisfactorily responded to my inquiries and made the necessary changes to the manuscript.